# Association between dietary calcium, potassium, and magnesium consumption and glaucoma

**Yin Zhang, Zhihua Zhao, Qingmin Ma, Kejun Li, Xiaobin Zhao, Zhiyang Jia** *

Department of ophthalmology, Hebei General Hospital, Shijiazhuang, Hebei, P.R. China

* zhiyangJhb@outlook.com

## Abstract

### Background

Calcium (Ca), potassium (K) and magnesium (Mg) may be involved in the occurrence and development of glaucoma by influencing the retinal oxidative stress and regulate blood pressure. However, epidemiological opinions on dietary intake of macroelement related to glaucoma are inconsistent. Herein, this study aims to explore the association between dietary Ca, K, and Mg consumption and glaucoma.

### Methods

Data of 7,042 adults aged ≥40 years old who received the glaucoma examinations were extracted from the National Health and Nutrition Examination Survey (NHANES) database from 2005 to 2008 in this cross-sectional study. Univariate and multivariate logistic regression were used to explore the association between dietary Ca, K, and Mg intake and glaucoma with the odd ratios (ORs) and 95% confidence intervals (CIs). We also investigated this relationship in individuals of different age, with/without hypertension and visual field defect (VFD).

### Results

There were 502 (8.11%) participants had glaucoma. After adjusted for covariables, we found that enough dietary Ca consumption was related to a decreased risk of glaucoma [OR = 0.59, 95%CI: (0.42–0.81)], whether in persons with/without hypertension (all $P<0.05$). In particular, dietary K intake may be a potential protect factor for glaucoma in non-hypertension populations [OR = 0.47, 95%CI: (0.22–0.99), $P = 0.049$]. Additionally, hypertension/non-hypertension persons who aged <65 years old or with/without VFD should all pay attention to the enough dietary supplement of Ca, K, and Mg according to their own circumstances.

**Data Availability Statement:** The datasets generated and/or analyzed during the current study are available in the NHANES database, https://wwwn.cdc.gov/nchs/nhanes/.

**Funding:** The external funding supported this study was 2018 Medical Science Research Key Project Plan of Hebei Province (20180052). There was no additional external funding received for this study.

**Competing interests:** The authors have declared that no competing interests exist.

## Conclusion

Enough dietary Ca, K, and Mg consumption may be potential protect factors of glaucoma that could provide some dietary reference for developing targeted glaucoma prevention and control measures.

## Introduction

Glaucoma is a chronic optic neuropathy characterized by the progressive loss of retinal ganglion cells [1]. The grave consequence of glaucoma is visual field loss and glaucoma is one of the leading causes of irreversible visual disability over the world that creates a significant burden on public health [2, 3]. The prevalence of glaucoma in people aged from 40 to 80 years old is approximated to be 3.5% globally [4]. With the growing number of older persons, it is projected that 111.8 million people will suffer from glaucoma in 2040 [5]. The pathophysiology of glaucoma is complex, and has not been completely understood. Elevated intraocular pressure (IOP) and/or impaired retinal blood flow may cause initial optic nerve damage [6]. Oxidative stress in the retina is also involved in glaucoma progression due to it is closely related to cell senescence, mitochondrial dysfunction, excitotoxicity, and neuroinflammation [7].

Calcium (Ca), potassium (K), and magnesium (Mg) are the essential macroelement to humans and the principal source is through diet [8]. Study have demonstrated the role of dietary Ca in modulating oxidative and inflammatory stress [9]. K is vasoactive and dietary supplementation of K can lower blood pressure in normal individuals [10]. Mg is involved in normal cell membrane function, energy metabolism, and synthesis of nucleic acids and exhibits beneficial effects through both neuronal and vascular mechanisms [11]. Limited epidemiological evidence suggest that dietary intake of Ca, K, and Mg may be associated with the risk of glaucoma, but the conclusions are still inconsistent. A population-based cross-sectional survey investigated the association between nutrient intake and primary open angle glaucoma (POAG) in Koreans, and showed that in normal body mass index (BMI) women, those with POAG had significantly lower intakes of Ca and K than their non-glaucoma counterparts [12]. Wang et al. [13] found that dietary Ca intake was associated with lower odds of glaucoma while supplemental consumption of its oxidants was related to the higher odds. Mozaffarieh et al. [14] considered that Mg can be used to improve blood flow in patients with glaucoma while Ramdas et al. [15] found Mg intakes seem to be higher in glaucoma patients than that in non-patients. Therefore, the effect of dietary Ca, K, and Mg intake on glaucoma needs to be further verified.

Additionally, recent studies remaindered that hypertension is associated with the increased eye pressure that could furthermore affect the occurrence and development of glaucoma [16–18]. Intraocular pressure elevation is among glaucoma's strongest risk factors [19]. And there were significant differences in oxidative stress and blood flow between patients with hypertension and that non-hypertensive [20]. However, there is currently a lack of research on the effect of dietary macroelement intake on glaucoma in hypertension or non-hypertension patients.

Herein, this study aims to explore the association between dietary Ca, K, and Mg consumption and the risk of glaucoma, and further investigate this relationship in hypertension and non-hypertension populations in order to provide some scientific reference for developing targeted glaucoma prevention and control measures.

## Methods

### Study design and population

This study is a cross-sectional study. The participants were selected from the National Health and Nutrition Examination Survey (NHANES) database from 2005 to 2008. NHANES, done by the National Center for Health Statistics (NCHS; Hyattsville, MD, USA), is a multipurpose research program to assess the health and nutritional status of population in the USA [21]. NHANES regularly collects data of approximately 5000 persons from 15 areas since 1999 that includes a household interview followed by a standardized physical examination in a mobile examination center. A stratified multistage sampling design with a weighting scheme based on the selection of counties, blocks, households, and persons within households is used by NHANES to represent the civilian, non-institutionalized US population and accurately estimate disease prevalence (https://www.cdc.gov/nchs/nhanes/index.htm).

A total of 7,042 adults were extracted since they aged ≥40 years old, received the glaucoma examinations and with complete information of 24-h dietary recall. The exclusion criteria were (1) with severe eye infections (according to the question 'Do you appear to have a severe eye infection in one or both eyes?'), (2) missing the information of study variables, and (3) reported an implausible energy intake (<600 or >4000 kcal/day for women and <800 or >6000 kcal/day for men) [22]. Finally, 6,189 eligible participants were enrolled in our study. The requirement of ethical approval for this was waived by the Institutional Review Board of Hebei General Hospital, because the data was accessed from NHANES (a publicly available database). The need for written informed consent was waived by the Institutional Review Board of Hebei General Hospital due to retrospective nature of the study.

### Assessment of dietary Ca, K, and Mg intake

Dietary Ca, K, and Mg intake was determined according to the two 24-h dietary recall interviews by the United States Department of Agriculture (USDA) automated multiple-pass method (AMPM) [23]. The first recall was conducted in person, and later recall was conducted 3–10 days after the first one via a phone call. In the NHANES 2005–2008, participants' nutrient content of foods consumption was determined by using relevant Food and Nutrient Database for Dietary Studies (FNDDS) for each NHANES release [24]. A dietary supplement questionnaire as part of the NHANES household interview, was used to assess the dietary supplements such as vitamins, minerals et. al. The consumption frequency, duration, and dosage were recorded for each of the dietary supplements over the prior 30 days that can be used for calculation of the average daily intake of nutrients [25]. Individual nutrient intakes from "Food Only" and from "Food Plus Supplement" were estimated using 24-h dietary recall in NHANES with the National Cancer Institute method and with the covariates including day of recall, weekday/weekend intake flag, and dietary supplement use (yes/no) flag [26].

We divided dietary Ca, K, and Mg intake levels into reached and below dietary recommendation levels according to their dietary recommendations respectively. The Ca estimated average requirement (EAR) is 800 mg/d [27]. The suggested K intake is 120 mmol/d (4.7 g/d) according to the American Heart Association issued guidelines [28]. Recommended dietary allowance (RDA) of Mg intake was developed by the Food and Nutrition Board (FNB) of the Institute of Medicine: 400 mg/d for male aged 19–30 years old, 420 mg/d for that aged ≥30 years old; 310 mg/d for female aged 19–30 years old, and 320 mg/d for that aged ≥30 years old [29].

## Variable collection

We collected the study variables from the database including age, gender, race, poverty-income ratio (PIR), education level, marital status, drinking and smoking status, physical activity, screening time, eye surgery for nearsightedness and cataracts, trouble seeing even with glass/contacts, hypertension, diabetes, usage of β-adrenergic blocking agents, body mass index (BMI), total cholesterol (TC), visual field defect (VFD) and energy intake.

## Outcome variable

The primary outcome of interest was the glaucoma. Using the question in the vision questionnaire of NHANES: "Have you ever been told by an eye doctor that you have glaucoma, sometimes called high pressure in your eyes?" to assess the prevalence of self-reported glaucoma. Retinal imaging and frequency doubling technology (FDT) visual field testing were offered to subjects as well.

In the NHANES from 2005 to 2008, retinal photographs were obtained from individuals aged ≥40 years old except those were unable to see light with both eyes open or had an eye infection. A Canon CR6-45NM nonmydriatic camera (Canon, Tokyo, Japan) was used to obtain the two 45 nonmydriatic digital images from both eyes (The first image is centered on the macula, and the second centered on the optic nerve). The images were initially graded using standardized methods for assessing vertical cup-to-disc ratio (vCDR) as a continuous numeric variable from 0.00 to 1.00 at the University of Wisconsin Fundus Photograph Reading Center. The vCDR asymmetry was calculated as the absolute value of the difference between the vCDR of the 2 eyes [30]. Later in 2012, the retinal images with a CDR greater than or equal to 0.6 were re-read with attention to features relevant to glaucoma by ophthalmologists based at Johns Hopkins University. Glaucoma in each eye was classified as "no, possible, probable, definite, or unable to assess." If at least two of three graders provided the same grade, and the third grader was within one level, then that grade was assigned to the image. If at least two of the graders did not agree, or if the third grader was off by two or more levels, then the image was re-read in the presence of all three graders in order to achieve consensus [31].

The FDT tests were performed by trained investigators in a dark room. FDT perimetry utilized the Humphrey Matrix Visual Field Instrument, and the assessment used the N-30-5 FDT screening protocol. Nineteen visual field locations of each eye were tested in the assessment until a response was received from each of the locations. Each of the participant's eyes were tested twice. A practice test was run prior to the actual test to determine whether the participant understood the test procedures. Identification of an abnormal visual field occurred when both the first and second test showed that at least two locations fell below a 1% threshold level, and at least one failed location was the same in both tests (2-2-1 algorithm). Three false-positive and blind-spot checks were carried out at random in order to check for the reliability of each test. The FDT result for each participant was subsequently categorized as normal, positive, insufficient, or unreliable [32].

An objective clinical definition of glaucoma is the Rotterdam criteria (using optic nerve appearance and VFDs). In brief, the diagnosis of glaucoma was made if any of the following were present: (1) having two or more abnormal points in at least one eye on the N30-5 FDT on two tests in the same eye, and (2) CDR in one eye or CDR asymmetry between eyes ≥97.5% of the normal NHANES population [33].

## Statistical analysis

The normal distribution data was described using mean ± standard deviation (mean ± SD) and t test for comparation between groups. Non-normal distribution was described by

median and quartiles [M (Q1, Q3)] and Mann-Whitney U rank test for the comparation. Categorical data were expressed as number with constituent ratio [N (%)] and chi-square test for the comparison.

Univariate logistic regression analysis was used to screen the covariates of glaucoma both in total participants and in that with/without hypertension. Then, we used the multivariate logistic regression analysis to explore the association between dietary Ca, K, and Mg consumption and the risk of glaucoma among the above populations. Model 1 adjusted for the energy intake. Model 2 of total individuals adjusted for age, gender, race, education level, marital status, smoking status, physical activity, screen time, eye surgery for cataracts, trouble seeing even with glass/contacts, diabetes, TC, usage of β-adrenergic blocking agents and energy intake. Model 2 of hypertension patients adjusted for age, race, education level, marital status, eye surgery for nearsightedness and cataracts, trouble seeing even with glass/contacts, diabetes and energy intake. Model 2 of participants without hypertension adjusted for age, education level, smoking status, marital status, physical activity, eye surgery for cataracts, trouble seeing even with glass/contacts, diabetes, TC, BMI and energy intake.

We also analyzed the relationship between Ca, K, and Mg consumption and glaucoma in subgroups of age and VFD. The evaluation index was odds ratios (ORs) with 95% confidence intervals (CIs). Two-sided $P<0.05$ is considered significant. Statistical analysis was performed using SAS 9.4 (SAS Institute, Cary, NC, USA). Missing data were showed in S1 Table, and there was no significant difference before and after imputation of missing data according to the sensitivity analysis (S2 Table).

## Results

### Characteristics of study population

Fig 1. was the flowchart of the participants screening. There are 7,042 individuals aged ≥40 years old and received the glaucoma examinations in NHANES from 2005 to 2008. We excluded those without information of the 24-h dietary recall (n = 631), had severe eye infection (n = 3), and had an implausible energy intake (n = 219). Finally, a total of 6,189 eligible participants were included for further analyses.

The characteristics of non-glaucoma persons and glaucoma patients were showed in Table 1. There are 502 (8.11%) participants suffered from glaucoma. Of the whole study population, the average age was 60.15±12.53 years old and 3,076 (49.70%) were male. The energy (1621.00 kcal vs. 1831.00 kcal), Ca (729.00 mg vs. 752.00 mg), K (2211.00 mg vs. 2448.00 mg), and Mg (238.00 mg vs. 262.00 mg) intakes were statistical differences between glaucoma group and non-glaucoma group. In addition, variables including PIR, education level, smoking status, physical activity, screen time, eye surgery for cataracts, trouble seeing even with glass/contacts, diabetes, hypertension, usage of β-adrenergic blocking agents, TC, and VFD are significantly different between the two groups as well (all $P<0.05$).

### Association between dietary macronutrient intake and glaucoma

We first screened for the covariates of glaucoma by univariate analysis (S3 Table). Age, race, PIR, education level, marital status, smoking status, physical activity, screen time, eye surgery for cataracts, trouble seeing even with glass/contacts, diabetes, BMI, TC, usage of β-adrenergic blocking agents and energy intake are significantly associated with glaucoma (all $P<0.05$). Then, after adjusting for the above covariates, we explored the association between dietary Ca, K, and Mg intake and the risk of glaucoma (Table 2). The results demonstrated that Ca consumption reached the recommended level is linked with a decreased risk of glaucoma [OR = 0.59, 95%CI: (0.42–0.81), $P = 0.002$].

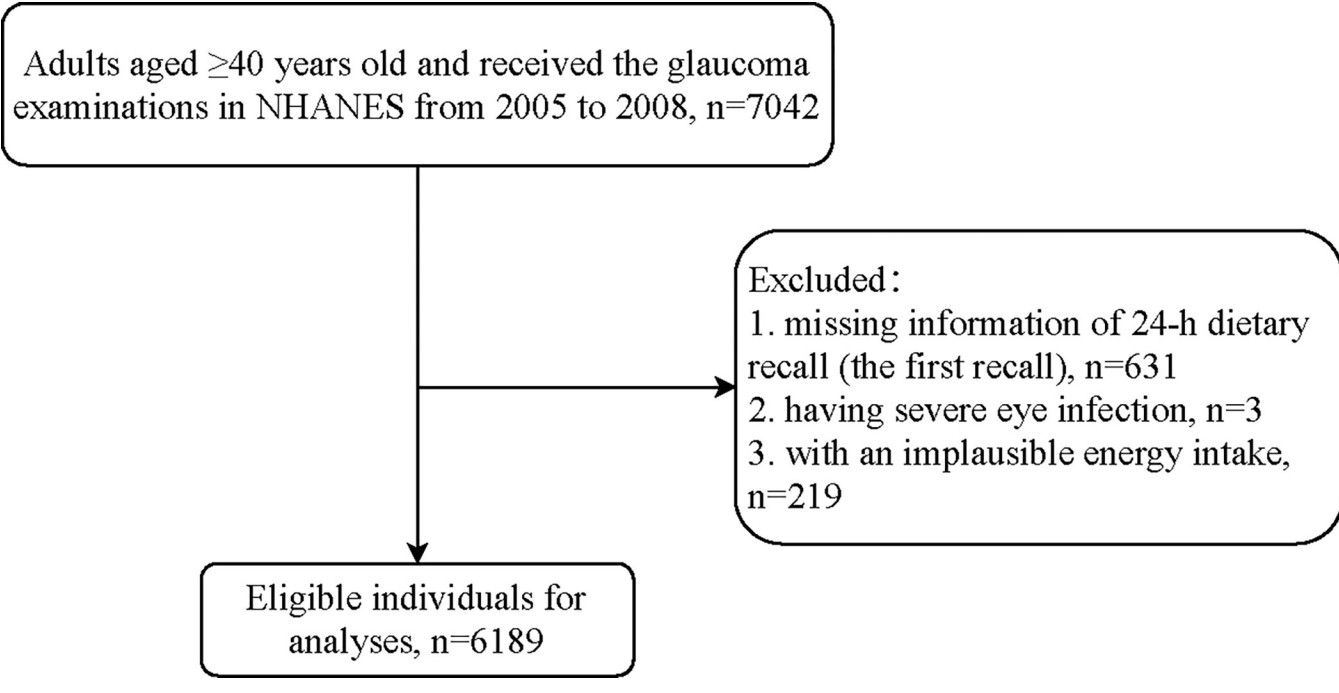

**Fig 1. The flowchart of the participants screening.**

### Effects of dietary Ca, K, and Mg consumption on glaucoma in hypertension/non-hypertension individuals

Since intraocular pressure elevation is one of the strongest glaucoma's risk factors that may be affect by hypertension [16, 19], we also explored the effect of dietary Ca, K, and Mg intake on glaucoma in non-hypertension and hypertension patients.

S4 Table shows the covariates of glaucoma while Table 3 is the multivariable analysis to explore the association between dietary Ca, K, and Mg intake and glaucoma in hypertension and non-hypertension individuals, respectively. It's worth noting that there are some differences of the covariates for glaucoma between hypertension group and non-hypertension group, including race, smoking status, physical activity, eye surgery for nearsightedness, BMI and TC. We further analyzed the relationship between dietary macronutrient intake and the risk of glaucoma in the two groups and found that dietary Ca intake level reached the recommendation linked with a decreased risk of glaucoma in both hypertension persons [OR = 0.67, 95%CI: (0.46–0.98), $P = 0.039$] and non-hypertension persons [OR = 0.41, 95%CI: (0.19–0.89), $P = 0.026$]. In particular, it seemed that enough dietary K consumption may decrease the risk of glaucoma in non-hypertension populations [OR = 0.47, 95%CI: (0.22–0.99), $P = 0.049$].

### Association between dietary macronutrient intake and glaucoma in age and VFD subgroups

According to our discovery that enough dietary macronutrient intake was related to glaucoma in hypertension/non-hypertension participants, subgroup analyses are conducted to explore this association in different populations.

In the populations aged <65 years old, enough dietary Ca intake was linked to a decrease risk of glaucoma in hypertension patients [OR = 0.32, 95%CI: (0.15–0.67), $P = 0.004$] and that without hypertension [OR = 0.32, 95%CI: (0.14–0.77), $P = 0.013$]. Differently, dietary K intake

**Table 1. Characteristic of the non-glaucoma persons and glaucoma patients.**

| Variables | Total (n = 6189) | Non-glaucoma (n = 5687) | Glaucoma (n = 502) | Statistics | P |
|---|---|---|---|---|---|
| Age, years, Mean±SD | 60.15 ± 12.53 | 59.45 ± 12.43 | 68.04 ± 10.91 | t = -16.71 | <0.001 |
| Gender, n (%) | | | | $\chi^2$ = 0.106 | 0.744 |
| Male | 3076 (49.70) | 2823 (49.64) | 253 (50.40) | | |
| Female | 3113 (50.30) | 2864 (50.36) | 249 (49.60) | | |
| Race, n (%) | | | | $\chi^2$ = 23.777 | <0.001 |
| Mexican American | 949 (15.33) | 893 (15.70) | 56 (11.16) | | |
| Non-Hispanic White | 3297 (53.27) | 3044 (53.53) | 253 (50.40) | | |
| Non-Hispanic Black | 1302 (21.04) | 1157 (20.34) | 145 (28.88) | | |
| Other Hispanic | 427 (6.90) | 393 (6.91) | 34 (6.77) | | |
| Other race (including multi-racial) | 214 (3.46) | 200 (3.52) | 14 (2.79) | | |
| PIR, M (Q₁, Q₃) | 2.39 (1.25, 4.48) | 2.42 (1.25, 4.55) | 2.06 (1.24, 3.92) | Z = -2.416 | 0.016 |
| Education level, n (%) | | | | $\chi^2$ = 15.486 | 0.004 |
| Less than 9th grade | 923 (14.91) | 828 (14.56) | 95 (18.92) | | |
| 9-11th grade (Includes 12th grade with no diploma) | 952 (15.38) | 867 (15.25) | 85 (16.93) | | |
| High school graduate/ GED or equivalent | 1526 (24.66) | 1400 (24.62) | 126 (25.10) | | |
| Some college or AA degree | 1552 (25.08) | 1428 (25.11) | 124 (24.70) | | |
| College graduate or above | 1236 (19.97) | 1164 (20.47) | 72 (14.34) | | |
| Marital status, n (%) | | | | $\chi^2$ = 74.133 | <0.001 |
| Married | 3634 (58.72) | 3381 (59.45) | 253 (50.40) | | |
| Widowed | 819 (13.23) | 691 (12.15) | 128 (25.50) | | |
| Divorced | 854 (13.80) | 792 (13.93) | 62 (12.35) | | |
| Separated | 221 (3.57) | 203 (3.57) | 18 (3.59) | | |
| Never married | 416 (6.72) | 386 (6.79) | 30 (5.98) | | |
| Living with partner | 245 (3.96) | 234 (4.11) | 11 (2.19) | | |
| Drinking status, n (%) | | | | $\chi^2$ = 1.091 | 0.580 |
| Frequently | 1363 (22.02) | 1261 (22.17) | 102 (20.32) | | |
| Occasionally | 2833 (45.77) | 2602 (45.75) | 231 (46.02) | | |
| Never | 1993 (32.20) | 1824 (32.07) | 169 (33.67) | | |
| Smoking status, n (%) | | | | $\chi^2$ = 30.339 | <0.001 |
| Yes | 1194 (19.29) | 1131 (19.89) | 63 (12.55) | | |
| No | 3002 (48.51) | 2774 (48.78) | 228 (45.42) | | |
| Quitted | 1993 (32.20) | 1782 (31.33) | 211 (42.03) | | |
| Physical activity, MET·min, n (%) | | | | $\chi^2$ = 9.993 | 0.002 |
| <450 | 3256 (52.61) | 2958 (52.01) | 298 (59.36) | | |
| ≥450 | 2933 (47.39) | 2729 (47.99) | 204 (40.64) | | |
| Screen time, n (%) | | | | $\chi^2$ = 7.935 | 0.019 |
| Long time | 1258 (20.33) | 1134 (19.94) | 124 (24.70) | | |
| Not long | 1416 (22.88) | 1318 (23.18) | 98 (19.52) | | |
| Unknown | 3515 (56.79) | 3235 (56.88) | 280 (55.78) | | |
| Eye surgery for nearsightedness, n (%) | | | | $\chi^2$ = 1.370 | 0.242 |
| Yes | 139 (2.25) | 124 (2.18) | 15 (2.99) | | |
| No | 6050 (97.75) | 5563 (97.82) | 487 (97.01) | | |
| Eye surgery for cataracts, n (%) | | | | $\chi^2$ = 152.590 | <0.001 |
| Yes | 840 (13.57) | 681 (11.97) | 159 (31.67) | | |
| No | 5349 (86.43) | 5006 (88.03) | 343 (68.33) | | |
| Trouble seeing even with glass/contacts, n (%) | | | | $\chi^2$ = 47.355 | <0.001 |
| Yes | 1542 (24.92) | 1353 (23.79) | 189 (37.65) | | |

*(Continued)*

**Table 1.** (Continued)

| Variables | Total (n = 6189) | Non-glaucoma (n = 5687) | Glaucoma (n = 502) | Statistics | P |
|---|---|---|---|---|---|
| No | 4647 (75.08) | 4334 (76.21) | 313 (62.35) | | |
| Diabetes, n (%) | | | | $\chi^2 = 53.105$ | <0.001 |
| No | 4707 (76.05) | 4392 (77.23) | 315 (62.75) | | |
| Yes | 1482 (23.95) | 1295 (22.77) | 187 (37.25) | | |
| Hypertension, n (%) | | | | $\chi^2 = 78.981$ | <0.001 |
| No | 2418 (39.07) | 2315 (40.71) | 103 (20.52) | | |
| Yes | 3771 (60.93) | 3372 (59.29) | 399 (79.48) | | |
| Usage of β-adrenergic blocking agents, n (%) | | | | $\chi^2 = 38.811$ | <0.001 |
| No | 5232 (84.54) | 4856 (85.39) | 376 (74.90) | | |
| Yes | 957 (15.46) | 831 (14.61) | 126 (25.10) | | |
| BMI, kg/m², Mean±SD | 29.19 ± 6.47 | 29.21 ± 6.50 | 29.02 ± 6.11 | t = 0.63 | 0.530 |
| TC, mg/dL, Mean±SD | 201.53 ± 42.51 | 202.13 ± 42.44 | 194.73 ± 42.70 | t = 3.75 | <0.001 |
| VFD, n (%) | | | | $\chi^2 = 294.749$ | <0.001 |
| No | 5660 (91.45) | 5304 (93.27) | 356 (70.92) | | |
| Yes | 529 (8.55) | 383 (6.73) | 146 (29.08) | | |
| Energy intake, kcal, M (Q₁, Q₃) | 1815.00 (1375.00, 2409.00) | 1831.00 (1383.00, 2428.00) | 1621.00 (1269.00, 2147.00) | Z = -5.721 | <0.001 |
| Ca intake, mg, M (Q₁, Q₃) | 750.00 (497.00, 1079.00) | 752.00 (500.00, 1085.00) | 729.00 (474.00, 1010.00) | Z = -2.518 | 0.012 |
| K intake, mg, M (Q₁, Q₃) | 2433.00 (1768.00, 3228.00) | 2448.00 (1778.00, 3248.00) | 2211.00 (1634.00, 2907.00) | Z = -4.189 | <0.001 |
| Mg intake, mg, M (Q₁, Q₃) | 261.00 (190.00, 350.00) | 262.00 (192.00, 352.00) | 238.00 (174.00, 309.00) | Z = -4.697 | <0.001 |
| Ca intake level, n (%) | | | | $\chi^2 = 18.624$ | <0.001 |
| Reached the dietary recommendation | 1556 (25.14) | 1470 (25.85) | 86 (17.13) | | |
| Below the dietary recommendation | 4633 (74.86) | 4217 (74.15) | 416 (82.87) | | |
| K intake level, n (%) | | | | $\chi^2 = 7.369$ | 0.007 |
| Reached the dietary recommendation | 1936 (31.28) | 1806 (31.76) | 130 (25.90) | | |
| Below the dietary recommendation | 4253 (68.72) | 3881 (68.24) | 372 (74.10) | | |
| Mg intake level, n (%) | | | | $\chi^2 = 7.064$ | 0.008 |
| Reached the dietary recommendation | 1284 (20.75) | 1203 (21.15) | 81 (16.14) | | |
| Below the dietary recommendation | 4905 (79.25) | 4484 (78.85) | 421 (83.86) | | |

t: test, Z: rank sum test

$\chi^2$: chi-square test

SD: standard deviation, PIR: poverty-income ratio, M: median, Q1:1st quartile, Q3:3rd quartile, MET: metabolic equivalent of energy, BMI: body mass index, TC: total cholesterol, VFD: visual field defect, Ca: calcium, K: potassium, Mg: magnesium.

reached the recommended level may be a potential protect factors for glaucoma in those who without hypertension [OR = 0.38, 95%CI: (0.15–0.92), P = 0.034] while that Mg for that glaucoma in hypertension patients [OR = 0.32, 95%CI: (0.13–0.77), P = 0.013] (Table 4).

For individuals without hypertension, dietary K [OR = 0.38, 95%CI: (0.17–0.89), P = 0.027] intake level reached the recommendation may associated with the decreased risk of glaucoma in that without VFD, while that enough Ca [OR = 0.05, 95%CI: (0.00–0.71), P = 0.028] intake may salutary for those with VFD (Table 5).

## Discussion

This study explored the association between dietary Ca, K, and Mg intake level and the risk of glaucoma, and further analyzed it in participants with or without hypertension. As the essential macronutrient for human body, enough level of dietary Ca intake was associated with the decreased risk of glaucoma, especially in those who with/without hypertension. Additionally,

**Table 2. Association between dietary Ca, K, and Mg intake and glaucoma.**

| Variables | Model 1 | | Model 2 | |
|---|---|---|---|---|
| | OR (95% CI) | *P* | OR (95% CI) | *P* |
| Ca intake level | | | | |
| Below the dietary recommendation | Ref | | Ref | |
| Reached the dietary recommendation | 0.52 (0.38–0.70) | <0.001 | 0.59 (0.42–0.81) | 0.002 |
| K intake level | | | | |
| Below the dietary recommendation | Ref | | Ref | |
| Reached the dietary recommendation | 0.88 (0.61–1.27) | 0.475 | 1.00 (0.68–1.47) | 1.000 |
| Mg intake level | | | | |
| Below the dietary recommendation | Ref | | Ref | |
| Reached the dietary recommendation | 0.81 (0.51–1.28) | 0.351 | 1.04 (0.65–1.65) | 0.878 |

Ca: calcium, K: potassium, Mg: magnesium, OR: odds ratio, CI: confidence interval, Ref: reference.

Model 1 adjusted for the energy intake; Model 2 adjusted for age, gender, race, education level, marital status, smoking status, physical activity, screen time, eye surgery for cataracts, trouble seeing even with glass/contacts, diabetes, TC, usage of β-adrenergic blocking agents and energy intake.

dietary K, Mg, and Ca intake level reached the recommendation may be potential protect factors for glaucoma in individuals aged <65 years old or with/without VFD.

Lee et al. [12] investigated the association between nutrient intake and POAG in Koreans and showed found that in normal BMI women, those with POAG had significantly lower intakes of Ca and K than their non-glaucoma counterparts. However, another study demonstrated that dietary Ca intake was linked to lower odds of glaucoma while its oxidants supplemental consumption had higher odds [13]. In our study, Ca consumption reached the recommended level was associated with a decreased risk of glaucoma. $Ca^{2+}$ is a second messenger that interacts with various cellular proteins and regulates multiple physiological processes that contribute to diseases including cancer, fibrosis, and glaucoma [34]. The intracellular concentration of $Ca^{2+}$ is low with a high inwardly directed concentration [35], and there are various pathways of cellular Ca regulation both extracellularly and intracellularly [36]. It is known

**Table 3. Association between dietary Ca, K, and Mg intake and glaucoma in hypertension/non-hypertension populations.**

| Variables | Non-hypertension | | Hypertension | |
|---|---|---|---|---|
| | OR (95% CI) | *P* | OR (95% CI) | *P* |
| Ca intake level | | | | |
| Below the dietary recommendation | Ref | | Ref | |
| Reached the dietary recommendation | 0.41 (0.19–0.89) | 0.026 | 0.67 (0.46–0.98) | 0.039 |
| K intake level | | | | |
| Below the dietary recommendation | Ref | | Ref | |
| Reached the dietary recommendation | 0.47 (0.22–0.99) | 0.049 | 1.04 (0.68–1.61) | 0.837 |
| Mg intake level | | | | |
| Below the dietary recommendation | Ref | | Ref | |
| Reached the dietary recommendation | 1.00 (0.49–2.04) | 0.997 | 0.77 (0.43–1.37) | 0.359 |

Ca: calcium, K: potassium, Mg: magnesium, OR: odds ratio, CI: confidence interval, Ref: reference.

Non-hypertension group: adjusted for age, education level, smoking status, marital status, physical activity, eye surgery for cataracts, trouble seeing even with glass/contacts, diabetes, TC, BMI and energy intake

Hypertension group: adjusted for age, race, education level, marital status, eye surgery for nearsightedness and cataracts, trouble seeing even with glass/contacts, diabetes and energy intake.

**Table 4. Association between dietary Ca, K, and Mg intake and glaucoma in age subgroup.**

| Variables | <65 years old | | ≥65 years old | |
|---|---|---|---|---|
| | OR (95% CI) | *P* | OR (95% CI) | *P* |
| **Non-hypertension** | | | | |
| Ca intake level | | | | |
| Below the dietary recommendation | Ref | | Ref | |
| Reached the dietary recommendation | 0.32 (0.14–0.77) | 0.013 | 0.51 (0.14–1.89) | 0.300 |
| K intake level | | | | |
| Below the dietary recommendation | Ref | | Ref | |
| Reached the dietary recommendation | 0.38 (0.15–0.92) | 0.034 | 0.75 (0.27–2.11) | 0.576 |
| Mg intake level | | | | |
| Below the dietary recommendation | Ref | | Ref | |
| Reached the dietary recommendation | 1.04 (0.45–2.39) | 0.929 | 0.69 (0.23–2.12) | 0.504 |
| **Hypertension** | | | | |
| Ca intake level | | | | |
| Below the dietary recommendation | Ref | | Ref | |
| Reached the dietary recommendation | 0.32 (0.15–0.67) | 0.004 | 0.92 (0.66–1.27) | 0.594 |
| K intake level | | | | |
| Below the dietary recommendation | Ref | | Ref | |
| Reached the dietary recommendation | 0.90 (0.38–2.16) | 0.807 | 1.03 (0.67–1.57) | 0.897 |
| Mg intake level | | | | |
| Below the dietary recommendation | Ref | | Ref | |
| Reached the dietary recommendation | 0.32 (0.13–0.77) | 0.013 | 1.12 (0.61–2.05) | 0.706 |

Ca: calcium, K: potassium, Mg: magnesium, OR: odds ratio, CI: confidence interval, Ref: reference.

Non-hypertension group: adjusted for age, education level, smoking status, marital status, physical activity, eye surgery for cataracts, trouble seeing even with glass/contacts, diabetes, TC, BMI and energy intake

Hypertension group: adjusted for age, race, education level, marital status, eye surgery for nearsightedness and cataracts, trouble seeing even with glass/contacts, diabetes and energy intake.

that aberrant $Ca^{2+}$ homeostasis, oxidative cell injury, and mitochondrial dysfunction are related to glaucoma [37–39]. In pathologic conditions, excessive $Ca^{2+}$ enters into the mitochondria and triggers Ca release that induce further $Ca^{2+}$ release [40]. ROS can damage these plasma membrane proteins that are able to sustain $Ca^{2+}$ concentration [41]. McElnea et al. [41] showed that ROS may decrease the organelles' ability to buffer $Ca^{2+}$ through impairing mitochondrial respiration and depolarizing the mitochondrial membrane thereby. Studies have indicated that increased levels of oxidative damage markers and reduced antioxidant potential in various ocular tissues of glaucoma patients [42, 43]. In addition, elevated intraocular pressure is considered a major risk factor for the incidence and/or progression of POAG [19]. Study demonstrated that Ca dysregulation results in the wild-type myocilin misfolding initially and may further contribute broadly to glaucoma-associated endoplasmic reticulum stress [44]. Therefore, we postulated that enough dietary Ca intake could supplement Ca loss in the cytoplasm after excessive $Ca^{2+}$ entered mitochondria during the progression of glaucoma to maintain $Ca^{2+}$ homeostasis in lamina cribrosa cells [34].

For individuals without hypertension, besides Ca, enough dietary K consumption may decrease the risk of glaucoma compare to hypertension patients. A cross-sectional and prospective cohort study on the association between serum ion levels with the risk of primary angle close glaucoma (PACG) showed that PACG patients had a significantly higher level of K compared with normal subjects [45]. A proper concentration of K in blood and extracellular

**Table 5. Association between dietary Ca, K, and Mg intake and glaucoma in VFD subgroup.**

| Variables | Non-VFD | | VFD | |
|---|---|---|---|---|
| | OR (95% CI) | *P* | OR (95% CI) | *P* |
| **Non-hypertension** | | | | |
| Ca intake level | | | | |
| Below the dietary recommendation | Ref | | Ref | |
| Reached the dietary recommendation | 0.47 (0.19–1.16) | 0.100 | 0.05 (0.00–0.71) | 0.028 |
| K intake level | | | | |
| Below the dietary recommendation | Ref | | Ref | |
| Reached the dietary recommendation | 0.38 (0.17–0.89) | 0.027 | 1.66 (0.22–12.65) | 0.609 |
| Mg intake level | | | | |
| Below the dietary recommendation | Ref | | Ref | |
| Reached the dietary recommendation | 0.91 (0.41–2.05) | 0.823 | 2.19 (0.29–16.56) | 0.432 |
| **Hypertension** | | | | |
| Ca intake level | | | | |
| Below the dietary recommendation | Ref | | Ref | |
| Reached the dietary recommendation | 0.59 (0.33–1.05) | 0.072 | 0.96 (0.44–2.07) | 0.910 |
| K intake level | | | | |
| Below the dietary recommendation | Ref | | Ref | |
| Reached the dietary recommendation | 0.95 (0.59–1.51) | 0.815 | 1.61 (0.66–3.90) | 0.284 |
| Mg intake level | | | | |
| Below the dietary recommendation | Ref | | Ref | |
| Reached the dietary recommendation | 0.65 (0.37–1.16) | 0.139 | 1.74 (0.59–5.14) | 0.308 |

Ca: calcium, K: potassium, Mg: magnesium, VFD: visual field defect, OR: odds ratio, CI: confidence interval, Ref: reference.

Non-hypertension group: adjusted for age, education level, smoking status, marital status, physical activity, eye surgery for cataracts, trouble seeing even with glass/contacts, diabetes, TC, BMI and energy intake

Hypertension group: adjusted for age, race, education level, marital status, eye surgery for nearsightedness and cataracts, trouble seeing even with glass/contacts, diabetes and energy intake.

fluid is critical to normal cellular function, and it is maintained in normal concentration with a limited fluctuation under normal conditions [46]. Difficulty in aqueous humor outflow through the trabecular meshwork plays a role in intraocular pressure elevation [47]. K dependent ATP ($K_{ATP}$) channels are existing in the trabecular meshwork, and their activation can increase outflow facility through the trabecular meshwork [48]. Stumpff et al. [49] demonstrated effluxion of K via maxi-K channels could result in trabecular meshwork relaxation. Study has shown that pharmacologic openers of functional K dependent ATP ($K_{ATP}$) channel lower the intraocular pressure in normotensive animal models and nonhuman primates [50, 51]. The alteration of K current may activate the cell volume regulation in trabecular meshwork that obstructed aqueous humor outflow and further resulted in increased intraocular pressure [52, 53]. Insufficient K intake increase the risk of hypertension [54]. A study found K inversely associated with systolic blood pressure and diastolic blood pressure [55]. Another US multi-center randomized controlled trial showed that a high level of K dietary intervention was linked to a significantly reduced mean of blood pressure [56]. Nevertheless, nutrients are intimately related and collectively control high blood pressure in the body. A K deficient diet promotes sodium (Na) retention that results in bloodstream increases and high blood pressure, while a high dietary K intake makes kidneys excrete more salt and water and then reduces blood pressure [57, 58]. Hypertension patients usually take hypotensive drugs or follow a blood-lowering dietary pattern to control K and Na balance. However, individuals that were

not diagnosed with hypertension may tend to have raised blood pressure or are more sensitive to the blood pressure lowering effects of K that need to be further explored.

In the populations aged <65 years old, enough dietary Ca and K intake was linked to a decrease risk of glaucoma. Age is an identified risk factor of glaucoma [59, 60]. Disruption of $Ca^{2+}$ homeostasis is a hallmark of ageing and disease [61], and ageing is an important contributor to the risk of hypertension in humans [62]. Long-lasting small or short-lasting large increases in intracellular $Ca^{2+}$ levels increase the risk of neurodegenerative diseases [63]. Prolonged increases in $Ca^{2+}$ may also result in mitochondrial $Ca^{2+}$ accumulation and dysfunction that lead to oxidative stress and inflammatory signaling [64, 65]. Study also found an increased all-cause mortality in older adults with the lowest levels of K intake [66]. Therefore, we speculated that the absorption efficiency of dietary supplement of Ca and K is not high enough for old persons to affect the occurrence and development of glaucoma. Additionally, higher levels of physical activities are related to a better state of health, whereas lower physical activity levels are related to the increased risk of chronic diseases and an increase in overall mortality [67]. Old persons usually with indifferent health that may affect the occurrence and development of glaucoma. Dietary Ca and K intake level reached the recommendation may associated with the decreased risk of glaucoma in participants with and without VFD, respectively. Progressive damage to the retinal ganglion cells (RGCs) that leads to axon loss and visual field alterations is a characteristic of glaucoma [68, 69]. Studies have shown a positive effect of Ca channel blockers on ocular blood flow and visual field in normotensive glaucoma (NTG) [70, 71]. Different dietary supplements should be used for glaucoma patients with different severity (with or without VFDs).

We collected the study population from NHANES database that the sample size was large and representative. To our knowledge, it is the first time to explore the association between dietary Ca, K, and Mg intake and glaucoma in individuals with hypertension or not. However, there are still some limitations in this study. This study is a retrospective study that no causal association could be established. A part of the glaucoma patients was diagnosed by self-reports. In addition, dietary review may be affected by some biases, but we have partly compensated for this limitation by excluding extreme values of energy intake and adjusting for them as covariates.

## Conclusion

Enough level of dietary Ca and K intake were related to glaucoma whether in persons with/without hypertension. Scientific reference of dietary macroelement intake for developing targeted glaucoma prevention and control measures should be considered respectively in high-risk populations.

## Supporting information

**S1 Checklist. STROBE Statement—checklist of items that should be included in reports of observational studies.**
(DOCX)

**S1 Table. Description of missing variables.**
(DOCX)

**S2 Table. Sensitivity analysis of characteristics of participants before and after imputation of missing data.**
(DOCX)

**S3 Table. Covariates of glaucoma.**
(DOCX)

**S4 Table. Covariates of glaucoma in hypertension/non-hypertension persons.**
(DOCX)

## Author Contributions

**Conceptualization:** Yin Zhang, Zhiyang Jia.

**Data curation:** Zhihua Zhao, Qingmin Ma, Kejun Li, Xiaobin Zhao.

**Formal analysis:** Zhihua Zhao, Qingmin Ma, Kejun Li, Xiaobin Zhao.

**Investigation:** Zhihua Zhao, Qingmin Ma, Kejun Li, Xiaobin Zhao.

**Methodology:** Zhihua Zhao, Qingmin Ma, Kejun Li, Xiaobin Zhao.

**Writing – original draft:** Yin Zhang.

**Writing – review & editing:** Yin Zhang, Zhiyang Jia.

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
