## [Decision Letter · Decision Letter 0]

21 Sep 2023

PONE-D-23-18980Association between dietary calcium, potassium, and magnesium consumption and glaucomaPLOS ONE

Dear Dr. Jia,

Thank you for submitting your manuscript to PLOS ONE. After careful consideration, we feel that it has merit but does not fully meet PLOS ONE’s publication criteria as it currently stands. Therefore, we invite you to submit a revised version of the manuscript that addresses the points raised during the review process.

We look forward to receiving your revised manuscript.

Kind regards,

Akram Belghith

Academic Editor

PLOS ONE

Journal Requirements:

"This study was supported by 2018 Medical Science Research Key Project Plan of Hebei Province (20180052)."

"This study was supported by 2018 Medical Science Research Key Project Plan of Hebei Province (20180052)."

"This study was supported by 2018 Medical Science Research Key Project Plan of Hebei Province (20180052)."

Reviewers' comments:

Reviewer's Responses to Questions

**Comments to the Author**

1. Is the manuscript technically sound, and do the data support the conclusions?

Reviewer #1: Yes

2. Has the statistical analysis been performed appropriately and rigorously? 

Reviewer #1: Yes

3. Have the authors made all data underlying the findings in their manuscript fully available?

Reviewer #1: Yes

4. Is the manuscript presented in an intelligible fashion and written in standard English?

Reviewer #1: Yes

5. Review Comments to the Author

Reviewer #1: Zhang et al. investigated the association between dietary Ca, K, and Mg consumption and glaucoma. I have only one minor question.

A Canon nonmydriatic camera was used to obtain the retinal image in the current study. Were all retinal images clear to assess vCDR? Older subjects often have lens opacity or miotic eye. These eye conditions often prevent to obtain clear retinal image.

6. PLOS authors have the option to publish the peer review history of their article (what does this mean?). If published, this will include your full peer review and any attached files.

Reviewer #1: No

---

## [Author Response · Author response to Decision Letter 0]

23 Sep 2023

Reviewer #1: Zhang et al. investigated the association between dietary Ca, K, and Mg consumption and glaucoma. I have only one minor question.

A Canon nonmydriatic camera was used to obtain the retinal image in the current study. Were all retinal images clear to assess vCDR? Older subjects often have lens opacity or miotic eye. These eye conditions often prevent to obtain clear retinal image.

Response: Thank you for your commont. The methods to assess vCDR were the standard methods provided in the NHANES database. The retinal image of eyes were evaluated by at least two of three professional eye doctors, and we belive that the eligible retinal images were all clear to assess vCDR.

---

## [Editor Report · Decision Letter 1]

2 Oct 2023

Association between dietary calcium, potassium, and magnesium consumption and glaucoma

PONE-D-23-18980R1

Dear Dr. Jia,

We’re pleased to inform you that your manuscript has been judged scientifically suitable for publication and will be formally accepted for publication once it meets all outstanding technical requirements.

Kind regards,

Akram Belghith

Academic Editor

PLOS ONE

---

## [Editor Report · Acceptance letter]

9 Oct 2023

PONE-D-23-18980R1 

Association between dietary calcium, potassium, and magnesium consumption and glaucoma 

Dear Dr. Jia:

I'm pleased to inform you that your manuscript has been deemed suitable for publication in PLOS ONE. Congratulations! Your manuscript is now with our production department. 

Kind regards, 

on behalf of

Dr. Akram Belghith 

Academic Editor

PLOS ONE